

**The Clam Before the Storm: A Meta-Analysis Showing the Effect of**
**Combined Climate Change Stressors on Bivalves**
Rachel A. Kruft Welton[1], George Hoppit[1], Daniela N. Schmidt[1], James D. Witts[1], Benjamin C. Moon[1]
[1]Bristol Palaeobiology Research Group, School of Earth Sciences, University of Bristol, Wills Memorial
Building, Queens Road, Bristol BS8 1RJ, UK
*Correspondence to*: george.hoppit@gmail.com
**Abstract.**
Impacts of a range of climate change on marine organisms have been analysed in laboratory and experimental
studies. The use of different taxonomic groupings, and assessment of different processes, though, makes
identifying overall trends challenging, and may mask phylogenetically different responses. Bivalve molluscs are
an ecologically and economically important data-rich clade, allowing for assessment of individual vulnerability
and across developmental stages. We use meta-analysis of 203 unique experimental setups to examine how
bivalve growth rates respond to increased water temperature, acidity, deoxygenation, changes to salinity, and
combinations of these drivers. Results show that anthropogenic climate change will affect different families of
bivalves disproportionally but almost unanimously negatively. Almost all drivers and their combinations have
significant negative effects on growth. Combined deoxygenation, acidification, and temperature shows the
largest negative effect size. Eggs/larval bivalves are more vulnerable overall than either juveniles or adults.
Infaunal taxa, including Tellinidae and Veneridae, appear more resistant to warming and oxygen reduction than
epifaunal or free-swimming taxa but this assessment is based on a small number of datapoints. The current focus
of experimental set-ups on commercially important taxa and families within a small range of habitats creates
gaps in understanding of global impacts on these economically important foundation organisms.



**1 Introduction**

Predictions of rising levels of atmospheric carbon dioxide indicate that the marine environment will significantly alter over the coming decades. Sea surface temperatures are projected to rise 2–4°C globally by the end of the century depending on region and emission scenario (IPCC, 2021). Higher latitudes will be exposed to more severe warming than the tropics (Meredith et al., 2019), resulting in sea ice and glacial melting, raising global sea levels, and increasing runoff and freshwater influx into marine settings (Lu et al., 2022). Ocean pH will decline by between 0.3–1 units by the end of the 21st century, with shallower waters expected to experience greater pH decreases than the open ocean (IPCC, 2021). Oxygen levels in the ocean are projected to decrease by up to 7% leading to an expansion of 'dead zones' (Breitburg, et al., 2018; Messié and Chavez, 2017; Schmidtko et al., 2017).

At the intersection of the marine and terrestrial realms, shallow marine and coastal settings typically exhibit large spatiotemporal variations in physicochemical conditions such as pH, oxygen, and temperature (e.g., Hoffmann et al., 2011). This variability is exacerbated by anthropogenic climate change leading to more frequent extreme climate events, as well as redistribution of upwelling zones, circulation and currents, and alterations to both the quantity and quality of terrestrial runoff (Sydeman et al., 2014). Therefore, environmental changes are projected to especially impact shallow marine and coastal habitats, which harbour socially and economically important ecosystems. Up to 40% of the world's population lives within 200 km of the coastline (Neumann et al., 2015), and an estimated 775 million people globally have high dependence on these systems and their services (Selig et al., 2019). Costal ecosystems are estimated to contribute more than 60% of the total economic value of the biosphere (Martínez et al., 2007) but organisms adapted to live in these systems are predicted to suffer large alterations in their population fitness in response to future climate change (Kroeker et al., 2013; Sampaio et al., 2021; Hoppit and Schmidt, 2022).

Bivalve molluscs (Class Bivalvia) are cornerstones of coastal and shallow marine ecosystems, with representatives in all marine biomes where they provide provisioning and regulatory services (Olivier et al., 2020; Carss et al., 2020; Vaughn and Hoellein, 2018). Bivalves are a key global food source, with production for human consumption growing from 51 thousand tonnes in 1950 to more than 12.4 million tonnes today (Smaal et al., 2019). As a result, bivalve aquaculture has an estimated global market value of $17.1 billion (van der Schatte Olivier et al., 2018).

Additionally, there is a growing awareness of the wider ecosystem benefits of bivalves as habitat formers. Their biogenic three-dimensional reefs formed by sessile epifaunal taxa such as oysters increase surrounding biodiversity (Fariñas-Franco and Roberts, 2014). Complex reef frameworks and individual bivalve shells in soft-substrate environments can act as microhabitats to other invertebrates through creation of new hard substrates and by altering local flow regimes or even temperature (McAfee et al., 2017). Filter-feeding bivalves can reduce pollution and clear large particulates out of the water column (Smyth et al., 2018). They enhance or change local marine productivity (Donnarumma et al., 2018; Strain et al., 2021) via selective removal of specific species of phytoplankton (Ward and Shumway, 2004). Deposition of bivalve faeces and pseudofaeces, along with burrowing activity of motile infaunal deposit or suspension feeders, can lead to changes in the local flux or enrichment of organic matter into the sediment and biogeochemical cycling (Smyth et al., 2013), and ultimately habitat suitability for other benthos.



Anthropogenic pressures including coastal development, overfishing, and pollution have resulted in
demonstrable decreases in global bivalve populations (e.g., De Groot, 1984; Baeta et al., 2014). Akin to other
calcifying invertebrate organisms, it has been hypothesised that the physiochemical changes resulting from
climate change will reduce bivalve growth, impair maintenance of shells (Maynou et al., 2020; Knights et al.,
2020), and disrupt larval settlement patterns and spawning (Bascur et al., 2020; Figueirodo et al., 2022). In
recognition of the environmental, social, and economic benefits bivalves produce, and the current and future
pressures they face, the group is a focus for conservation efforts (zu Ermgassen et al., 2020; Buelow and
Waltham, 2020, Gagnon et al., 2020). However, despite extensive study there remain significant gaps in our
understanding of their response to climate change across different bivalve families.
Current understanding of how bivalves will respond to various climate change stressors is based on field studies
and lab-derived experimental data focused largely on ocean acidification and response to warming, generally
observing negative responses (e.g., Beukema et al., 2009; van Colen et al., 2012; Addino et al., 2019; Eymann et
al., 2020). Synthesis work through meta-analysis supports the notion that bivalves will respond negatively to
climate change (Kroeker et al., 2013; Harvey et al., 2013; Sampaio et al., 2021; Hoppit and Schmidt, 2022;
Leung et al., 2022). These studies have shown that the synergistic effects of ocean acidification, ocean warming,
and an increase in hypoxic events decrease the growth rates of calcifying marine organisms (Kroeker et al.,
2010; Maynou et al., 2020; Knights et al., 2020; Sampaio et al., 2021). However, these analyses have been
conducted at high taxonomic rankings, e.g., examining changes at phylum level thereby they risk averaging
differential outcomes at finer taxonomic resolution. High level analyses can be difficult to interpret due to
clumping diverse responses into generalized trends (Helmuth et al., 2005). Organisms experience disparate
responses to environmental drivers based on local phenotypic expression (Dong et al., 2017) and environmental
influences (Genner et al., 2010) resulting, for example, in species-specific mortality risk to extreme heat based
on local microclimates and adaptation (Montalto et al., 2016). Therefore, our current understanding of how
bivalves respond to climate change based on broad scale synthesis work might not capture the granularity and
diversity of responses this group exhibits.
We aimed to fill this gap by employing a meta-analysis methodology to explore the effects of marine climate
stressors, and combinations thereof, on bivalve growth at both the whole-group and family levels. We address
the question of whether a negative response to climate change is intrinsic to the group or driven by specific taxa.
We focused on studies that emphasize bivalve growth rates; a commonly studied trait that offers insight into
organism vulnerability to answer how these growth rates are impacted by climate stressors, and whether
different families or developmental stages are more sensitive to climate stressors than others. Additionally, we
examine the range of experimental work assessing bivalve sensitivity to climate change to understand which
families are most represented. We hypothesise that a focus on commercially important bivalve taxa may be
creating a likely bias in observations.



## 2 Methods

### 2.1 Study selection criteria

Primary literature focusing on bivalve growth was identified using Web of Science Core collection. The keywords used were "bivalve", "Bivalvia", "meta-analysis", "acidification", "pH", "hypercapnia", "ocean change" "temperature", "salinity", "oxygen", "hypoxia", "anoxia", and combinations thereof. Articles collected ranged from 1997–2020. Articles were screened initially through title relevance, then abstract content, and finally full-text content (Fig. 1), from which individual experimental set-ups were extracted. Article lists from previous meta-analyses with similar scope (Kroeker et al., 2013; Harvey et al., 2013; Sampaio et al., 2021) were additionally consulted to identify material missed from initial search strings. For a list of included articles used for analysis please consult 'Data availability' section.

[Figure 1] PRISMA flow diagram of screening process for the present study following recommended guidelines (Page et al., 2021). Relevant articles on Bivalvia growth experiments were identified from the Web of Science Core Collection using a series of keywords (see main text). Screening resulted in the identification of 79 relevant articles with 203 experimental set-ups that were included in our meta-analysis.

We included articles with lab-based studies that focused on direct measurements of Bivalvia growth including length, mass, condition index, or shell thickness. Proxies for growth were excluded, such as scope for growth or RNA production, as these introduce additional uncertainties and variability to the growth signal and were not directly comparable to absolute measures of growth. Only studies where the bivalves were fed and studies on larvae that develop without feeding were included, as nutrient intake has a significant impact on growth (Norkko et al., 2005; Thomsen et al., 2013; Ballesta-Artero et al., 2018). Study sample size, mean growth value of both control and treatment groups, and indication of the variation of growth values (confidence intervals, standard error, and standard deviation) were extracted from articles. Absolute values were used, as percentage data could not be combined with absolute measurements within the Metafor package. Data were extracted directly from result text, tables, or supplementary data when possible. Data from figures was collected using WebPlotDigitizer v. 4.4 (Rohatgi, 2022). Control values for climate stressors for each article were based on authors' determination of control conditions. Climate stressor values were based on realistic end of century projections based on author's determination for that experimental setup or study location. The phylogeny and column chart (Fig. 2) were plotted using R v. 4.1.0 (R Core Team, 2021) and the packages ggplot2 v. 3.3.5 (Wickham, 2016), ggtree v. 3.2.1 (Yu et al., 2017), ape v. 5.6.1 (Paradis and Schliep, 2019), and patchwork v. 1.1.1 (Pederson, 2020). The topology is taken from the time-scaled 'budding II' family-level phylogeny of Crouch et al. (2021).

[Figure 2] Experimental representation of 18 Bivalvia families in 203 unique experimental setups from 79 relevant articles found in Web of Science Core Collection. **A**, time-scaled 'budding II' phylogeny of extant Bivalvia from Crouch et al. (2021). The root age is 485.4 Ma. **B**, number of experiments representing each extant family.

### 2.2 Statistical analysis

We preformed meta-analysis on the impacts of climate stressors on the growth of Bivalvia at whole-class and family levels following methods described in Hoppit and Schmidt (2022). Stressors identified from the included



experiments are water oxygen depletion ($O_2$), increased acidity (decreased pH), salinity decrease (S), and
temperature increase (T), and combinations of these stressors (indicated as, e.g., $O_2$ + pH) (Figs 3–5; Table 2).
Stressor effects could be synergistic (additive) or antagonistic (dampening) (*sensu* Harvey et al., 2013), or
dominated by one stressor (unaffected by changes in another stressor). Additionally, we separated out the effect
sizes of these stressors on different growth stages (egg/larva, juvenile, adult) for the entire class Bivalvia.
Analyses used R v. 4.0.3 (R Core Team, 2020) and the package Metafor version 3.0-2 (Viechtbauer, 2010).
Metafor function escalc was used to calculate effect size and sampling variance. We chose Log Response Ratio
(LnRR; the natural log of the response ratio) as the measure of effect size to measure the proportion of change
between the mean of the treatment and control responses to experimental intervention. An effect size of zero
corresponds to a statistically insignificant effect. Multivariate meta-analytical models (function rma.mv) were
used to calculate mean effect sizes of climate stressor impacts on bivalve growth rates for three subsets of data:
all bivalves pooled, different developmental stages, and families with sufficient sample sizes (n ≥ 7). Significant
results were identified when model 95% confidence intervals did not overlap zero effect size. Models used
random intercepts for articles and species intercepts for each treatment to compensate for similarities introduced
by studies, as data originating from the same experimental setup or from the same species are assumed to be
more likely similar than data from different articles or species. Residual heterogeneity (QE), calculated as part
of the meta-analytical models, was used to determine whether additional study moderators not considered might
be influencing study results (Hedges and Olkin, 1985).
Publication bias was tested using Egger's regression test. Following Habeck and Schultz (2015), function
rma.mv was extended using the square root of effect size variance in the model moderator variables to conduct a
regression test. Egger's regression test looks at the symmetry of the data published and determines whether there
are statistically 'missing' studies within the spread of the papers published (Egger et al., 1997). We used meta-
regression to determine whether published results had changed over the 25 years from which studies had been
collated, using study year as a moderator variable. This would indicate whether increasingly detailed knowledge
has altered the overall picture with regards to the effect of each climate change stressor.

## 158    3 Results

Our literature search produced the most detailed examination of bivalve growth rates under climate stressors to
date. We identified 79 studies with 203 unique experiments meeting the criteria, comprising 18 families and 37
species (Figs 1, 2; Table 1). Sampling of families was highly uneven: Mytilidae make up 36% of the
experiments and 81% of the experiments include just four families: Mytilidae, Ostreidae, Pectinidae, and
Veneridae; including Pinnidae and Tellinidae increases the total to 88% (Table 1).
We find consistent and significant negative effects of all single stressors and most combinations acting on the
entire class Bivalvia, in agreement with previous meta-analyses (Fig. 3; Table 2). At the class level, many
combinations of stressors increase the negative effect on growth in a synergistic way (Fig. 3; Table 2). For
example, pH and $O_2$ treatments are greater in combination than either alone, as were salinity + temperature and
pH + temperature. The effect of pH + salinity is intermediate between that of the two single stressors,
dampening the salinity effect, while $O_2$ + temperature causes a smaller effect than either single stressor. The





combination of three stressors, $O_2$ + pH + temperature, causes the strongest negative effect size to both
individual stressors and any combinations. While low heterogeneity is preferable in terms of data validity it is
rarely achievable in environmental meta-analyses. Therefore, the significant heterogeneity in the data is
expected given it is drawn from so many disparate studies: QE = 300509.7155, df = 148, $P < 0.0001$.
[Figure 3] Effect size (log-response ratio, LnRR) for individual and combined effects of temperature (T), acidity
(pH), oxygenation ($O_2$), and salinity (S) as stressors on bivalve growth rates for all Bivalvia. Points represent
mean effect size with error bars indicating 95% confidence intervals. Numbers indicate number of included
experiments. Significance is indicated with asterisks: * $P < 0.05$, ** $P < 0.01$, *** $P < 0.001$.
[Table 1] Representation of bivalve families across 203 experimental studies included in this meta-analysis.
[Table 2] Single and combined stressor effect sizes from a meta-analysis of 203 experimental set-ups (log-
response ratio, LnRR). Significance is indicated with asterisks: * $P < 0.05$, ** $P < 0.01$, *** $P < 0.001$.
Thirty-one of the 203 experimental set-ups involve adult Bivalvia, 14 on unspecified ages/stages, 45 on eggs/
larvae, and the remaining 113 focused on juvenile stages. Separating by growth stage shows that the
combination of pH and $O_2$ stressors causes significantly negative effect size at all points in the life cycle (Fig.
4). Salinity is not a significant stressor for larval or juvenile bivalves but causes a significant reduction in growth
in adults. Juveniles show responses to most stressors, whereas egg/larvae and adult bivalves have much smaller
sample sizes, and do not show significant effect size responses across the stressors.
Families do not all respond in the same way as the whole class Bivalvia, and stressors affect different families in
unexpected ways (Fig. 5). Mytilidae, Ostreidae, and Pectinidae (67% of experiments) respond with negative
effect sizes for all individual stressors (Fig. 5A–C). Pinnidae show positive responses for single stressors
temperature and pH, but negative when in combination (Fig. 5D). Tellinidae show positive responses for oxygen
and $O_2$ + pH (Fig. 5E). Veneridae (14% of experiments) show mixed results with significant negative effect
sizes of salinity, pH + S, $O_2$ + pH, and $O_2$ + pH + T, but strong positive responses to temperature and $O_2$ + T
(Fig. 5F).
[Figure 4] Effect size (log-response ratio, LnRR) for individual and combined effects of oxygenation ($O_2$),
acidity (pH), salinity (S), and temperature (T) as stressors on Bivalvia growth rates at different life stages
(egg/larval, juvenile, adult). Points represent mean effect size with error bars indicating 95% confidence
intervals. Numbers indicate number of included experiments. Significance is indicated with asterisks: * $P <$
0.05, ** $P < 0.01$, *** $P < 0.001$.
[Figure 5] Effect size (log-response ratio, LnRR) for individual and combined effects of oxygenation ($O_2$),
acidity (pH), salinity (S), and temperature (T) as stressors on Bivalvia growth rates separated by family. **A**,
Mytilidae. **B**, Ostreidae. **C**, Pectinidae. **D**, Pinnidae. **E**, Tellinidae. **F**, Veneridae. Points represent mean effect
size with error bars indicating 95% confidence intervals. Numbers indicate number of included experiments.
Significance is indicated with asterisks: * $P < 0.05$, ** $P < 0.01$, *** $P < 0.001$.
Egger's regression test showed highly significant ($P < 0.001$) results for every stressor, indicating publications
with significant results are published more often than would be expected by chance, suggesting negative



observations are less frequently reported (see Appendix A; Table A1). Meta-regression analysis of publication
by year and stressor showed that no individual stressor is changing in effect size signal through time, showing
consistency in publication findings over the years (see Appendix B; Fig. B1 and Table B1).

**4 Discussion**

The impact of individual and combined climate stressors on growth rates of bivalve molluscs in our study
concurs with previous meta-analyses on marine calcifying invertebrates. The findings re-iterate that as a group,
bivalves are highly vulnerable to conditions projected to occur under future climate change. Our analysis
demonstrates that increased incidences of deoxygenation, pH decrease, as well as changes to temperature and
salinity in nearshore marine environments in the future will inhibit the growth of bivalves. However, by
focusing specifically on bivalves and separating out both family-level response and different life stages, we
build upon previous synthesis work by revealing previously unappreciated complexity in responses. Effects of
climate change for this group will additionally to the physico-chemical environment depend on the varied
ecological and taxonomic makeup of specific habitats and will vary across growth stages which exploit the
habitat differently as plankton to settling as benthos. We also highlight that numerous biases exist in currently
available studies (taxonomic, ecological, geographic) which currently hinder upscaling of individual bivalve
responses to a true global picture.

**4.1 Climate change stressors will negatively impact bivalve growth**

Our findings clearly show that growth rates in Bivalvia are negatively affected by climate stressors (Fig. 3).
Previous meta-analyses that incorporated bivalves did not focus on the group specifically but include them
alongside numerous other taxa (Harvey et al., 2013; Kroeker et al., 2013; Sampaio et al., 2021). These analyses
which average over a wide range of taxa found little evidence for significant effect sizes except in a few single
stressors (pH and temperature: Harvey et al., 2013; hypoxia: Sampaio et al., 2021). Unsurprisingly, the effect of
temperature on bivalve growth is the most studied stressor in the experiments included in our meta-analysis (35
experiments: Fig. 3). This bias is likely because temperature-altering experiments require less complex
equipment and sensors than pH or oxygen manipulation, have been performed over decades, and target the most
obvious effect of climatic change i.e., global warming. Inclusion of a substantially greater number of previous
experiments within our meta-analysis (over 200 specific bivalve growth observations versus 45, 46, and 34
Mollusca for Kroker et al. (2013), Harvey et al. (2013), and Sampaio et al. (2021) respectively) confirms that all
single climate stressors show significant negative effect sizes in bivalves (Fig. 3). Our analysis also shows that
in many cases this effect prevails when individual growth stages (Fig. 4) and the families containing the largest
number of experiments or observational data (Table 1; Fig. 5) are examined separately.
An important result is the identification of synergistic, additive, and antagonistic effects between different
stressors which in all cases in our analysis increase the negative response (Gobler et al., 2014; Stevens and
Gobler, 2018) (Table 2). For example, we identify significant negative effect sizes for $O_2$ + pH, and temperature
+ salinity when analysing overall bivalve responses (Fig. 3). The combination $O_2$ + pH has a stronger negative
effect size than either oxygen or pH individually in all analyses (Figs. 3-5). Decreases in pH restrict growth via
restricting availability of $CO_3^{2-}$ and increasing $HCO_3^-$ ions making shell building more metabolically expensive





and increasing shell dissolution (Ivanina et al., 2013; Byrne and Fitzer, 2019). Internal tissues also require
buffering against pH changes, incurring a further metabolic cost (Byrne and Fitzer, 2019). Marine
deoxygenation impacts metabolism, reducing an organism's ability to respond to these increased metabolic
requirements of shell generation and tissue buffering under a more acidic environment. Therefore, the increased
impact from combining these two stressors confirms our physiological understanding of the organism (Pörtner
and Farrell, 2008).
Mean effect sizes for each climate stressor differ between families within Bivalvia. Consequently, the effects of
climate change on this group will be habitat dependant and alter the taxonomic composition of coastal
ecosystems. The four most investigated families in our dataset (Mytilidae, Osteridae, Pectinidae, and Veneridae)
exhibit consistent negative growth responses to climate stressors (Fig. 5). Exceptions exist for oxygen and
temperature changes for Mytilidae and Veneridae and temperature increase for Veneridae where we find mixed
responses. However, pH causes antagonistic decreases in growth rate across these main families (Fig. 5),
suggesting that any temperature-driven growth increases are unlikely to occur under future projected conditions.

**4.2 Different bivalve life stages and ecologies show distinct responses to climate stressors**

Climate change will be acting on each part of the development of an organism. In bivalves, these different life
stages have different habitats and mobility from free swimming larvae to sessile adults. Our results on how
different bivalve life stages are affected by a range of climate stressors generally confirm previous meta-
analyses. Egg/larval bivalve growth rates display the largest number of negative responses to single climate
stressors, followed by juveniles, with adults showing more mixed responses (Fig. 4). This suggests early life
stages are the most vulnerable to a specific set of stressors and that the threat diminishes as organisms mature,
supporting analyses by Sampaio et al. (2021) and Kroeker et al. (2013) which focused primarily on the impacts
of ocean acidification. It is important to note though that the earlier developmental stages are more mobile and
hence more able to relocate their niche to track their environmental needs.
Combined climate stressors (e.g. pH + temperature, $O_2$ + pH, salinity + temperature) showed negative responses
across all growth stages impacts on growth throughout ontogeny and different stages of life history. Our
findings oppose those of Harvey et al. (2013) who suggested limited variation in organism growth responses
exists between life stages exposed to individual and synergistic ocean acidification and warming. Their data
were pooled from multiple phyla not specific taxonomic groups reiterating the need to avoid too much pooling
and averaging in meta-analysis.
Most studies in our dataset, especially those on the families that dominate our meta-analysis (Fig. 4), are
focused on larval or juvenile stages. This likely explains some of the negative impacts on growth as for example
shell mineralogy influences the impact of pH changes. Amorphous calcium carbonate secreted by larval
bivalves is 50-times more susceptible to dissolution than either calcite or aragonite (Bressan et al., 2014).
Changes to body size or decreased shell thickness could increase vulnerability to predation (Sadler et al., 2018).
Non-significant effects of lowered pH alone in adult bivalves likely result from more diverse shell mineralogy
(Weiss et al., 2002), the effects of a more robust adult shell (Beadman et al., 2003), or shelf formation of adults
from a high $p$CO$_2$ low pH micro-environment quite different to the surrounding seawater (Thomsen et al., 2010;
Hiebenthal et al., 2013). The adult's lifestyle, which includes for some taxa exposure to air and/or closed valves



while respiring naturally results in high variability of pH in the calcifying fluid and therefore the pH changes in
the experiments may be resulted in relatively less stress compared to earlier developmental stages. Most of the
adult experiments included in our meta-analysis were on aragonitic individuals or on mixed aragonitic-calcitic
Mytilidae and Pectinidae. Only one study (Lemasson et al., 2018) included two genera of adult oysters (Family
Ostreidae) which construct their shells primarily from calcite (Stenzel, 1963), a more stable carbonate
polymorph. Our results suggest adults have an increased susceptibility to salinity changes when compared to
juvenile and egg/larval stages, suggesting habitats projected to experience decreased salinity due to increased
seasonal runoff or enhanced evaporation in a restricted setting (Robins et al., 2016) will become challenging for
adult bivalves.
Increased sensitivity to climate stressors at different life stages has implications for bivalve aquaculture and
conservation effort (Smaal et al., 2019). Hence an increased frequency of these conditions will be disruptive to
lifecycles in some taxa. Decreased growth rates in larval and juvenile stages might impact population
recruitment by limiting the number of individuals surviving to adulthood. Settlement efficacy will affect
repopulation success, following disturbance (Gagnon, et al., 2021). Aquaculture and fisheries will need to
account for these increased vulnerabilities and adapt culturing strategies to compensate for the negative growth
impacts of climate change.
Efforts to restore historical bivalve populations such as oysters in Chesapeake Bay (Bersoza Hernández et al.,
2018) and Europe (Sas et al., 2020) will need to consider how climate stressors will impact population
dynamics. Restoration projects often transplant adult or juvenile species into new environments (Johnson et al.,
2019); and our findings suggest transplanting adults might be a preferable strategy given the lower impact of
climate stressors at this developmental stage.

**4.3 Consideration of habitat and ecology in the context of climate change**

Many species belonging to the families Mytilidae, Osteridae, and Veneridae occur in intertidal habitats which
experience frequent fluctuations in oxygen, acidity, and temperature and has been hypothesised to provide some
species with an innate ability to resist or mitigate the effects of future environmental change (Gazeau et al.,
2013; Zhang et al., 2020). Furthermore, this high variability may include environments overlapping with those
replicated in some of the experimental setups. Species can in natural environments evade some stressors vai
behavioural thermoregulation, for example mobile infaunal bivalves have been shown to migrate to more
offshore habitats, or burrowing deeper into the sediment (Domínguez et al., 2020; Dominguez et al., 2021).
Negative growth responses though generally repeat across the taxa in our dataset irrespective of habitat. An
intertidal habitat or preference for marine or brackish water does not appear to alter observed growth responses
in the experimental setting to accumulated climate stressors, as we find consistent decreases in growth rates, and
commonly subtidal, epifaunal bivalves (such as many Pectinidae) also exhibiting significant negative responses
(Aguirre-Velarde et al., 2019; Maynou et al., 2020). Interpreting the effects of ecology on our results is
complicated by the previously mentioned dominance of studies focused on juvenile and early growth stages;
many bivalves feature a veliger or early larval stage that live in and can tolerate quite different environmental



conditions to those of later stages of life history (i.e., pelagic, free-swimming larvae vs infaunal or benthic
attached lifestyles for juveniles and adults) (Waldbusser et al., 2013).
The ability to evade will depend on the lifestyle and habitat. Most experiments in our dataset are suspension
feeding taxa with an epifaunal habitat. The investigated bivalves are free swimming (Pectinidae), cemented to
substrates or form biogenic reefs (Ostreidae), or use byssal threads to anchor in sediments or attach to hard
substrates (Pinnidae, Mytilidae, some Pectinidae). There is much lower representation in our dataset of infaunal
or burrowing taxa which may also include deposit feeders (e.g., families Tellinidae, Veneridae). Our data
suggest overwhelmingly negative impacts on growth of all stressors for epifaunal or free-swimming suspension
feeding taxa (families Mytilidae, Ostreidae, Pectinidae in Fig. 5).
Tellinidae and Veneridae show more varied responses to temperature, pH, and $O_2$ depletion. These taxa are
active infaunal burrowers in soft sediment, which can often be undersaturated with respect to calcium carbonate
as well as oxygen-limited (Green et al., 2013; Stevens and Gobler, 2018) suggesting some resilience to these
conditions. Both semi-infaunal Pinnidae and infaunal Veneridae also show significant positive effect sizes in
response to warming (Fig. 5). It has been hypothesised that an infaunal habitat may reduce immediate
susceptibility of bivalves to warming, as the substrate may act as a thermal refugia (Zhou et al., 2022). However,
interpreting the general role of tiering is complicated by the currently small number of experiments or
observations on infaunal taxa, further highlighting the need for additional data on the effects of environmental
stressors on the growth of burrowing bivalves and those from a wider range of specific shallow marine habitats.
**4.4 Experimental studies of bivalve response are biased by commercially important taxa**
Our meta-analysis clearly reveals that available data on bivalve growth responses to climate stressors contain a
number of biases. The majority of experimental set-ups are limited to a few families (e.g. Mytilidae [73],
Ostreidae [31], Pectinidae [32], Veneridae [28]) (Fig. 2; Table 1), with a focus on epifaunal (Mytilidae,
Pectinidae) or reef-building taxa (Ostreidae) that inhabit both intertidal and subtidal zones, and limited number
of infaunal (Veneridae) or semi-infaunl (Pinnidae) taxa. This bias is likely due to the commercial importance of
these families and individual species within them for aquaculture and common ecosystem services (e.g., van der
Schatte Olivier et al., 2020), as well as ease of access specimens. Many bivalve specimens were sourced from
commercial aquaculture facilities. A number of families included in our meta-analysis are represented only by
individual experiments: for example, Dreissenidae, Hitellidae, Mesodesmatidae, Myidae and Pharidae.
Comparison of the number of experiments vs. bivalve phylogeny shows that entire families have no documented
experimental or observational work investigating climate stressor impacts on growth (Fig. 2).
Our findings are also geographically biased towards the global north. Most studies clearly gathered organisms
and data from the coasts of the USA, Europe, or China, resulting in significant portions of the global ocean like
the Caribbean or African coasts being unrepresented in these data. While our meta-analysis focused specifically
on bivalve growth, this result emphasizes the unevenness of experimental research into this group. If this
disparity of understanding is not rectified then implementing effective climate change adaptation and mitigation
strategies and upscaling these results to ecosystem-scale changes are challenging.



While our experimental sample is larger than previous meta-analyses, these biases also leave much uncertainty
about how responses will scale up from commercially important species to other, rarely studied groups of
bivalves, which while of lesser importance for aquaculture or commercial exploitation, may act as keystone
species within fragile marine ecosystems. This further limits the quality and quantity of available information
that conservationists and stakeholders need to develop strategies to safeguard marine social-ecological systems.
Given our findings overwhelmingly suggest that bivalves as a group (Figs 3, 4) and common families (Fig. 5)
will likely experience decreased growth rates under protected projected end-of-century conditions, how likely is
it that families or species with no current experimental observations will also follow this trend? Additional
experimental and observational work on specific bivalve species and families is urgently required which would
greatly assist in developing conservation strategies for this important group of marine calcifiers.
**5 Conclusions**
Reduced growth rates predicted by our meta-analysis have important implications for population stability in
these commercially important keystone marine taxa, as well as for guiding future conservation and mitigation
efforts. Our meta-analysis concludes that growth rates of bivalve molluscs significantly decrease when exposed
to climate stressors. We demonstrate that synergistic combinations of stressors (e.g., effects of combined
temperature + $O_2$ + pH change) cause greater reductions in bivalve growth then individual stressors. This result
is true for bivalves overall, and when separating out by growth stage in the most commonly studied bivalve
families (Ostreidae, Mytilidae, Pectinidae, Veneridae).
Eggs/larval stages are significantly more susceptible to reduced growth then other developmental stages. The
potential effects on recruitment, as well as settlement and recovery after disturbance. has important implications
for conservation or transplant efforts, suggesting a renewed focus on transplanting adult specimens rather than
larvae/juveniles should be examined.
Epifaunal filter feeders, such as Ostreidae and Mytilidae, had mostly negative growth responses to
environmental stressors. In contrast, infaunal and semi-infaunal suspension or deposit feeding bivalves,
Veneridae, Tellinidae, and Pinnidae showed more mixed or even positive growth response under higher
temperatures, suggesting that burrowing or buried taxa may be buffered from some changes. However, these
data are based on a small number of studies, and these families still showed negative growth effects with other
stressors and combinations of stressors.
We highlight that available data on bivalve response to climate stressors has large biases towards early or
juvenile growth stages, commercially important species from the global north, and that a large proportion of
bivalve families lack any rigorous experimental or observational data. Regardless of these biases, our results
suggest that climate change will greatly affect marine bivalves, interacting with other stresses these organisms
already face.





**Appendices**

**Appendix A**

**Table A1.** Publication bias results of Egger's regression test.

For all stressors: df=195

|  | estimate | se | pval | tval | ci.lb | ci.ub |
|---|---|---|---|---|---|---|
| sqrt(vi):StressorO2 | -5.5054 | 0.367 | <.0001 | -15.0026 | -6.2292 | -4.7816 |
| sqrt(vi):StressorpH | -1.5811 | 0.3003 | <.0001 | -5.2654 | -2.1733 | -0.9888 |
| sqrt(vi):StressorpH and O2 | -10.929 | 0.3705 | <.0001 | -29.4969 | -11.6597 | -10.1982 |
| sqrt(vi):StressorpH and temperature | -7.2009 | 0.3165 | <.0001 | -22.7541 | -7.8251 | -6.5767 |
| sqrt(vi):Stressorsalinity | -1.5428 | 0.7805 | 0.0495 | -1.9765 | -3.0823 | -0.0033 |
| sqrt(vi):Stressorsalinity and pH | -10.1106 | 0.9205 | <.0001 | -10.9841 | -11.9261 | -8.2951 |
| sqrt(vi):Stressortemperature | -0.8807 | 0.3563 | 0.0143 | -2.4717 | -1.5834 | -0.1779 |
| sqrt(vi):Stressortemperature and O2 | -1.0071 | 0.5775 | 0.0828 | -1.7439 | -2.1462 | 0.1319 |
| sqrt(vi):Stressortemperature and pH and O2 | -9.7482 | 0.5994 | <.0001 | -16.2629 | -10.9304 | -8.5659 |
| sqrt(vi):Stressortemperature and salinity | -4.2012 | 0.869 | <.0001 | -4.8344 | -5.9153 | -2.4872 |



**Appendix B.**
**Fig. B1** Change of effect sizes of 203 experimental setups on Bivalvia growth through time from 1997 to 2020.
**A**, acidity (pH). **B**, temperature. **C**, deoxygenation. **D**, salinity. Each point shows the effect size against the data
set publication year. Point size indicates the experiment contribution weight to the linear model. Each plot
shows the regression of effect size against publication year with the 95% confidence interval shaded. All
regression analyses show no significant change during this period.

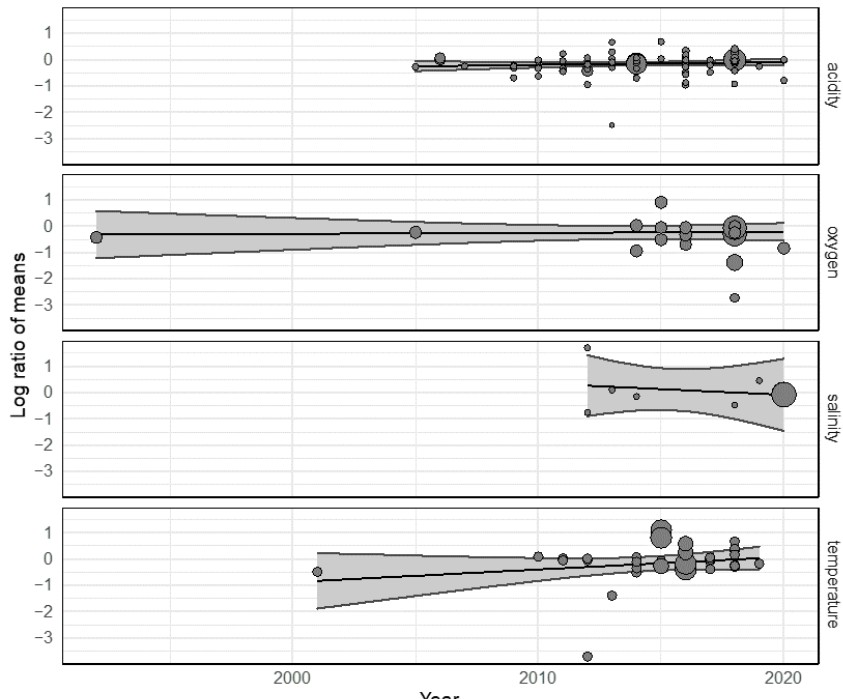



**Table B1.** Regression analysis of publications by stressor per year.

| Acid | estimate | se | tval | df | pval | ci.lb | ci.ub |
|---|---|---|---|---|---|---|---|
| intrcpt | -21.2396 | 19.6612 | -1.0803 | 88 | 0.2830 | -60.3120 | 17.8328 |
| Year | 0.0105 | 0.0098 | 1.0725 | 88 | 0.2864 | -0.0089 | 0.0299 |
| Temp | | | | | | | |
| intrcpt | -96.5465 | 74.7711 | -1.2912 | 33 | 0.2056 | -248.6695 | 55.5764 |
| Year | 0.0478 | 0.0371 | 1.2888 | 33 | 0.2064 | -0.0277 | 0.1233 |
| Oxygen | | | | | | | |



| intrcpt | -7.5967 | 36.9562 | -0.2056 | 16 | 0.8397 | -85.9404 | 70.7470 |
| Year | 0.0037 | 0.0183 | 0.1990 | 16 | 0.8447 | -0.0352 | 0.0425 |
| Salinity | | | | | | | |
| intrcpt | 87.7248 | 203.0242 | 0.4321 | 6 | 0.6808 | -409.0574 | 584.5071 |
| Year | -0.0435 | 0.1007 | -0.4315 | 6 | 0.6812 | -0.2900 | 0.2030 |


Signif. codes:  0 '***' 0.001 '**' 0.01 '*' 0.05 '.' 0.1 ' ' 1
























**Code availability**

Code used for analyses available at https://github.com/georgehoppit/Bivalve-meta-analysis

**Data availability**

Data used for analyses available at https://github.com/georgehoppit/Bivalve-meta-analysis

**Author contributions**

Conceptualization: GH, BCM; data curation, formal analysis, investigation, methodology: RKW, GH; resources, software, validation: RKW, GH, BCM, JDW; supervision: GH, JDW, BCM; visualization: RKW, GH, BCM; writing – original draft: RKW; writing – review & editing: RKW, GH, DNS, BCM, JDW. (https://credit.niso.org)

**Competing interests**

The authors declare that they have no conflict of interest.

**Acknowledgements**

BCM is supported by European Research Grant 788203 (INNOVATION) to Prof. Michael Benton (Bristol). GH is supported by NERC Scholarship Grant number NE/L002434/1. DNS is supported by the Leverhulme Trust grant RF-2021-489\4.



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



**Figure 1.** PRISMA flow diagram of screening process for the present study following recommended guidelines (Page et al., 2021). Relevant articles on Bivalvia growth experiments were identified from the Web of Science Core Collection using a series of keywords (see main text). Screening resulted in the identification of 79 relevant articles with 203 experimental set-ups that were included in our meta-analysis.

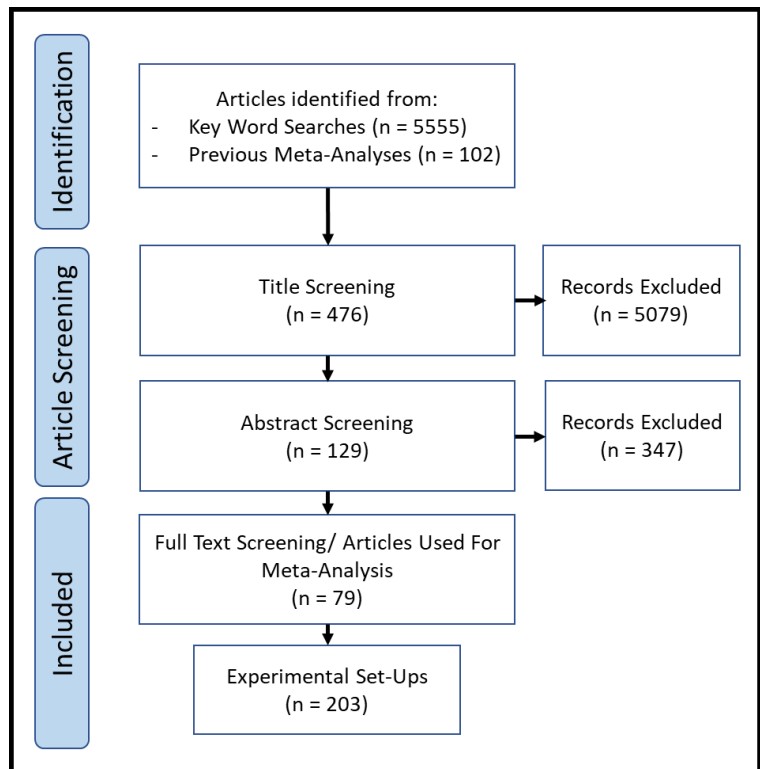



**Figure 2.** Experimental representation of 18 Bivalvia families in 203 unique experimental setups from 79
relevant articles found in Web of Science Core Collection. **A**, time-scaled 'budding II' phylogeny of
extant Bivalvia from Crouch et al. (2021). The root age is 485.4 Ma. **B**, number of experiments
representing each extant family.

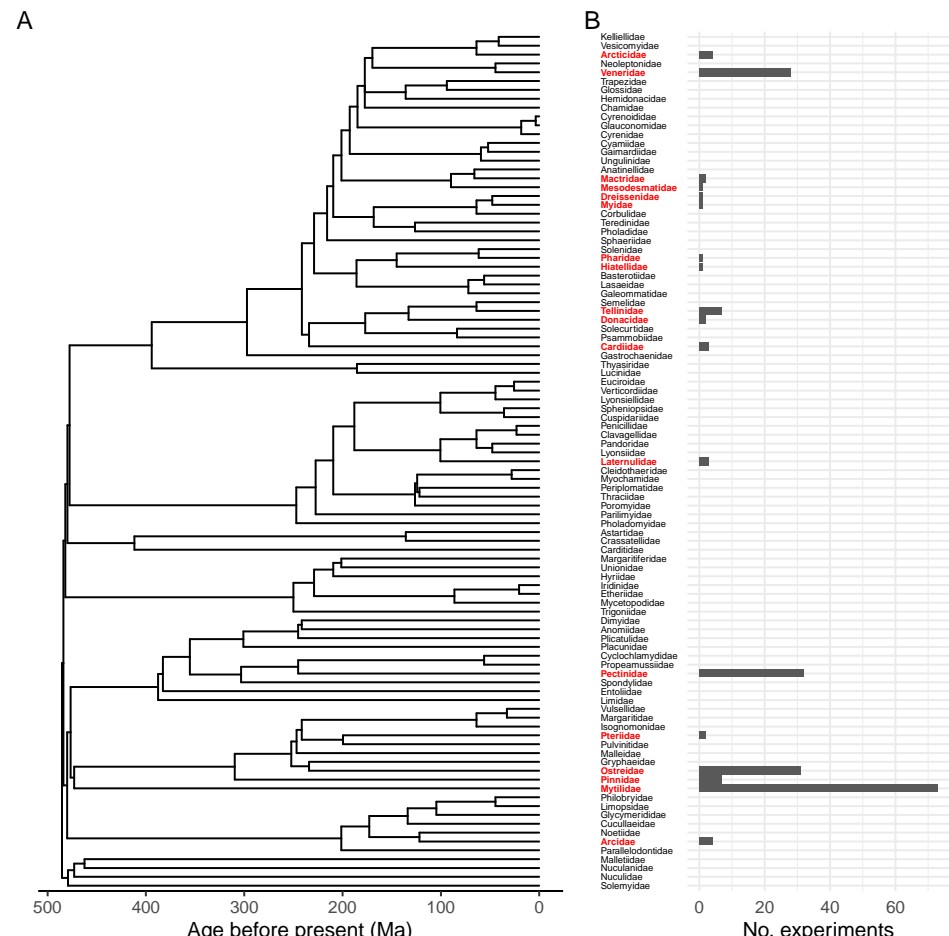





**Figure 3.** Effect size (log-response ratio, LnRR) for individual and combined effects of temperature (T), acidity (pH), oxygenation ($O_2$), and salinity (S) as stressors on bivalve growth rates. **A**, for all Bivalvia. **B**, for Bivalvia excluding Veneridae. Points represent mean effect size with error bars indicating 95% confidence intervals. Numbers indicate number of included experiments. Significance is indicated with asterisks: * $P < 0.05$, ** $P < 0.01$, *** $P < 0.001$.

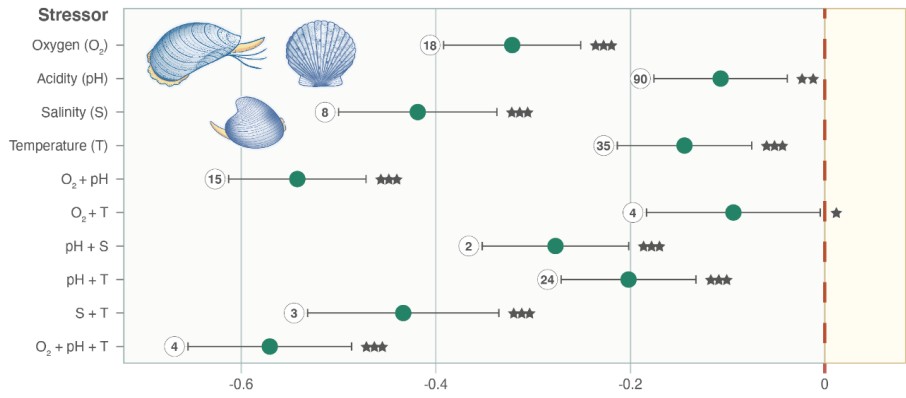



**Figure 4.** Effect size (log-response ratio, LnRR) for individual and combined effects of oxygenation (O₂), acidity (pH), salinity (S), and temperature (T) as stressors on Bivalvia growth rates at different life stages (egg/larval, juvenile, adult). Points represent mean effect size with error bars indicating 95% confidence intervals. Numbers indicate number of included experiments. Significance is indicated with asterisks: * $P < 0.05$, ** $P < 0.01$, *** $P < 0.001$.

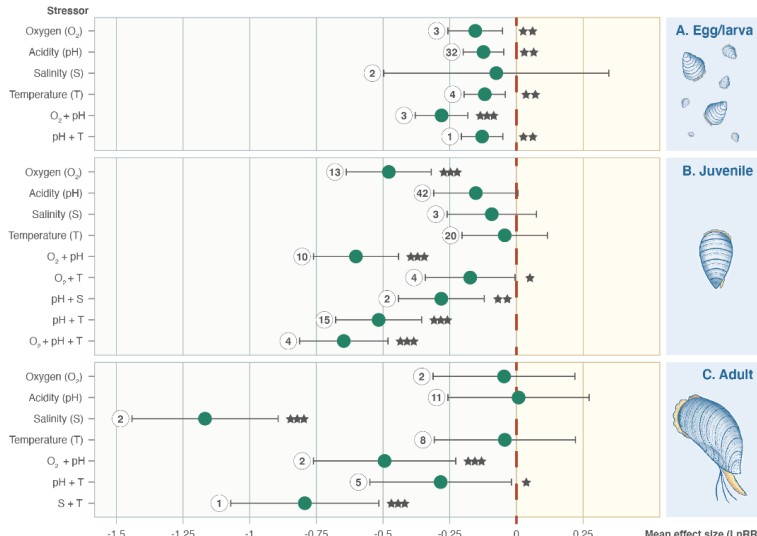



**Figure 5.** Effect size (log-response ratio, LnRR) for individual and combined effects of oxygenation
(O₂), acidity (pH), salinity (S), and temperature (T) as stressors on Bivalvia growth rates separated
by family. **A**, Mytilidae. **B**, Ostreidae. **C**, Pectinidae. **D**, Pinnidae. **E**, Tellinidae. **F**, Veneridae.
Points represent mean effect size with error bars indicating 95% confidence intervals. Numbers
indicate number of included experiments. Significance is indicated with asterisks: * $P < 0.05$, ** $P <$
0.01, *** $P < 0.001$.

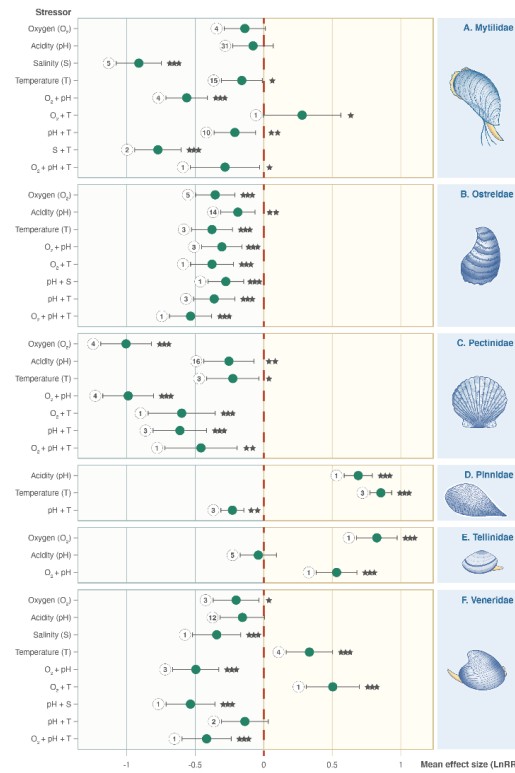









**Table 1.** Representation of bivalve families across 203 experimental studies included in this meta-analysis.

| Family | Number of experimental set-ups |
|---|---|
| Arcidae | 4 |
| Arcticidae | 4 |
| Cardiidae | 3 |
| Dinacidae | 2 |
| Dreissenidae | 1 |
| Hiatellidae | 1 |
| Laternulidae | 3 |
| Mactridae | 2 |
| Mesodesmatidae | 1 |
| Myidae | 1 |
| Mytilidae | 73 |
| Ostreidae | 31 |
| Pectinidae | 32 |
| Pharidae | 1 |
| Pinnidae | 7 |
| Pteriidae | 2 |
| Tellinidae | 7 |
| Veneridae | 28 |





**Table 2.** Single and combined stressor effect sizes from a meta-analysis of 203 experimental set-ups (log-
787          response ratio, LnRR). Significance is indicated with asterisks: * $P < 0.05$, ** $P < 0.01$, *** $P < 0.001$.

| Stressor | Sample size | Mean effect size ($R$) | 95% confidence interval lower | upper | $P$-value |
|---|---|---|---|---|---|
| Oxygenation ($O_2$) | 18 | -0.3214 | -0.3916 | -0.2513 | <.0001 |
| Acidity (pH) | 90 | -0.1077 | -0.1762 | -0.0392 | 0.0022 |
| Salinity (S) | 8 | -0.4184 | -0.4997 | -0.3372 | <.0001 |
| Temperature (T) | 35 | -0.1445 | -0.2135 | -0.0756 | <.0001 |
| | | | | | |
| $O_2$ + pH | 15 | -0.5421 | -0.6126 | -0.4716 | <.0001 |
| $O_2$ + T | 4 | -0.0944 | -0.1836 | -0.0052 | 0.0382 |
| pH + S | 2 | -0.2771 | -0.3522 | -0.2019 | <.0001 |
| pH + T | 24 | -0.2021 | -0.2712 | -0.1330 | <.0001 |
| S + T | 3 | -0.4335 | -0.5316 | -0.3354 | <.0001 |
| | | | | | |
| $O_2$ + pH + T | 4 | -0.5703 | -0.6542 | -0.4864 | <.0001 |


