# Peer review of "The Clam Before the Storm: A Meta-Analysis Showing the Effect of"

_EGUsphere, 2023_

## Author Response (AR1)

REVIEWER ONE

"This is a valuable, well-written meta analysis of climate change impacts on bivalves, focusing largely on experimental work.   The authors do a good job of explaining what is new about their work  - illustrating differential responses at the family-level, among life stages and lifestyles, complexity of responses, and revealing biases in study taxa.  There is thoughtful discussion of life stage differences, mechanisms underlying impacts, and implications for aquaculture, fisheries, conservation and restoration.   For those not working with bivalves, it might be interesting for the authors to speculate in the discussion which of their findings might be general across classes (or phyla) , and which might be specific to bivalves (or possibly other calcifying taxa) and why."

**We thank Prof Levin for their time and positive comments. We appreciate they feel our manuscript adds value to the wider field. We are delighted they felt our manuscript would prompt other researchers to consider the taxonomic granularity of their respective study groups considering our findings and interpretations.**

"It took me a while to find the link to the 79 articles used in the analysis.  Perhaps indicate this earlier  in the methods.  I could not find the original data derived from these articles that formed the basis of the analysis.  These should be provided to the reader as a supplement or separate database. I think this is standard practice for this type of work."

**We have moved the link to the GitHub page that contains the data and supporting code to the top of the methods section to improve the viability of our data.**

"Below are comments, questions and suggestions intended to strengthen the presentation."

"Introduction"

"Line 25-32.  These average environmental changes do not really tell the whole story since the  regional variability is huge (e.g. some areas have lost 40% of their oxygen)."

**We have added 'with some areas, such as the Gulf of Mexico, already suffering frequent, severe deoxygenation events (Breitburg, et al., 2018).' to highlight regional variability**

"Methods – It is hard from the text to evaluate what magnitude of stressors were applied (under specific RCP or SSP scenarios?)  Were studies limited to coastal species?  Perhaps provide a species list - or did I miss this?"

**The text has been updated to illustrate we extracted data from studies which produced realistic end-of-century projections for ocean conditions. The data for specific species is available in our accompanying data.**

"Line 90   By selecting one trait (growth) but not including survivorship – how many studies were excluded?  There are a number of papers that might have contributed to this study if survivorship were addressed.   I recognize growth is sublethal and the survivorship is lethal ; would you expect results to be very different?"

**The reviewer raises an interesting point about whether survivorship would create different results, however this is beyond the scope of our study.**

**The number of excluded papers is shown in figure 1, where we excluded 347 paper not meeting the requirements for our analysis. As we focussed on growth not lethal conditions, we cannot quantify what the difference would without repeating the screening, as this information was not recorded due to the different focus.**

**Many studies do report survivorship often as a percentage, but these frequently do not contain all the information needed to preform meta-analysis. The meta-analysis approach required studies reporting sample mean, variance, and sample size in order to calculate the effect sizes that fed into the meta-analytical models. Thus, studies just reporting survivorship as a percentage cannot be used in the same approach, and hence part of our decision to exclude this trait response and focus on growth.**

"Line 100 Why was deoxygenation not a search term?  Did this come up under oxygen or not?"

**"deoxygenation" was a permutation of oxygen that was covered by our search terms when identifying articles. We have included 'deoxygenation' in our list of search terms.**

"Line 131 preformed or performed?"

**Edit: performed**

"Natural variability in seawater associated with diel cycles, upwelling/seasonality , respiration as well as warming effects on gas solubility causes stressors to change together. So warmer temperatures are usually associated with lower oxygen, nighttime respiration draws down pH and O2 together – etc. Thus bivalves would be expected to be adapted to related changes. Perhaps discuss how this does or does not manifest in your results?"

**We have added this point to the introduction to introduce the information early on (see comment above).**

**Our methods have been updated to that data come from experiments that maintained constant conditions, and updated our discussion to reflect the reviewer's valuable point and how this is not accounted for in our results but is an important consideration.**

"Line 187-88  The results are stated to be unexpected.  This means that there were specific expectations.  Could these be presented as hypotheses? Is it the different responses among families that were unexpected?  Or the contrasting response to combinations of treatments?"

**We have edited the text throughout to better phrase the point we wished to convey in this line: Based on available data, families do not all respond in the same way as the whole class Bivalvia.**

"This is stylistic – but for sentences where you give the statistical result at the start of the sentence and the scientific meaning at the end of the sentence – I suggest reversing these and leading with the science, which is what the reader cares most about (e.g., lines 204-208)."

**We have considered this throughout the text. For the specific example, we edited: Publications with significant results are published more often than would be expected by chance, suggesting negative observations are less frequently reported (see Appendix A; Table A1), as shown by highly significant Egger's regression test (P < 0.001) results for every stressor.**

"Line 307 – typo in via"

**Fixed**

REVIEWER TWO

"The manuscript of Kruft Welton et al. reports on a meta-analysis focused on the effects of climate change stressors (pH, O2, temperature and salinity) on bivalves. Many papers have been published in the last decade showing that these stressors (isolated and/or combined) have significant effects on these species. The present study and manuscript follow a bunch of meta-analyses not necessarily focusing on this group but that performed separated analyses on them: Kroecker et al. (2013) showed that based on 45 datasets bivalve growth is significantly impacted by pH,

Harvey et al. (2013) showed that the combination of warming and acidification is synergistic, Sampaio et al. (2021) focused on the effects of pH, temperature and O2, the most recent Leung et al. (2022) restricted their study on pH but on a very comprehensive database. The main "new" approach here is to perform meta-analyses on different families of bivalves that were studied so far. Although I believe that identifying some bias in the literature towards economically important species is important (but that a narrative review would have done), I do not recommend considering this manuscript for publication for the three following reasons:"

**We thank the reviewer for taking time to consider our manuscript. We disagree with the conclusion that our manuscript should not be considered for publication as supported by the positive comments from reviewer 1. We present the most comprehensive known overview of datasets extracted from the literature recording the response of bivalves to climate change. Our manuscript explores a greater range and combination of climate change stressors than other known meta-analyses. Further, the value added by performing our meta-analysis at a family level highlights the variability of responses in what is often considered a well-studied group. Thereby our results show that previous work suggesting that bivalves/ molluscs respond poorly to climate change is not an assessment of bivalves overall but in reality only 3-4 families driving the signal. Our approach encourages future work to explore higher taxonomic granularity, as well as address the other biases we identify, which will support conservation and management strategies in light of climate change by emphasising disparity in organismal responses, which is needed by practitioners.**

"Apart from separating by families (but I will come back on that after), I do not see what is the real novelty presented here, I am sorry but we already knew that bivalves are impacted by climate changes, that larval stages are certainly the most sensitive stage etc..."

**We have updated the manuscript throughout to better highlight the novelty of our work, and the value it adds to understandings climate change. This includes rewriting the abstract, updating the introduction, rephrasing points in our discussion, and shorting and rewriting our conclusions to better convey our message.**

**We highlight our analysis explores a much greater combination of climate change stressors then past work, specifically 10 stressor combinations, while**

**the largest previous work in terms of stressors (Sampaio et al., 2021) explored 4. We consider this a strong development toward assessing the complexity needed and facilitated by a growing number of multi-stressor experiments.**

**We present the first known meta-analysis for marine climate change which explores the response of an individual group, and thereby identify that even a very well-studied group such as bivalves has large gaps in taxonomic understandings of how individual organisms respond to climate change. Our analysis was not intended to specifically point out that bivalves respond negatively to climate change, but highlight the taxonomic, geographic, and ecological variability experienced by the group, and provide an overview of the current state of experimental data. Therefore our findings should encourage a new approach to meta-analysis by moving towards more differentiated taxonomic understanding of a group's responses to future conditions, as well as hopefully stimulate additional data collection. Only this approach is able to show the level of variability that can be present within individual groups while still providing a summative response above the individual species.**

"I have strong concerns about the use of meta-analyses when protocols are so different from each other (level of perturbation applied, exposure time, amount of food provided (artificial vs natural, 1 phytoplankton species or a cocktail of species etc...), parameter or process measured: tissue growth, shell growth, dry mass etc...). Nothing is said here. Even the most important to me, the level of perturbation: L121. "Climate stressor values were based on realistic end of century projections based on author's determination for that experimental setup or study location." In the data table (thanks for sharing), pH offsets can be as low as -0.8 (maybe more, as the table is difficult to read), or as small as -0.2 (actually where do come from the 0.3-1 pH units mentioned in the introduction for the end-century projected changes in pH based on the IPCC?). It is very hard to see what is the range of O2 changes considered here, since units are different in the table and not easy to read, are they realistic? For temperature, the same, what is the range of offsets considered?"

"Were these factors considered as constant or were they varying? Very important to consider in the coastal ocean where strong dynamics are very common, and placing an organism under stable conditions is already a source of changes compared to their natural environment."

"What about salinity? When you mention pH, you refer to acidification (decrease in pH), when you mention T, you refer to warming (increase in T), when you mention O2, you refer to deoxygenation (right?), when you mention salinity, it is not clear if you refer to decrease or increase in salinity. Actually, from the data table, I see that both low salinity and high salinity effects are mixed together, isn't that weird?"

**The experimental literature we collected our database from is disparate, with papers having a wide range of experimental protocols to collect their data. Such a summative analysis across setups is the power of meta-analysis and the basis of all published analyses of this type. While we see the concern, the reviewer is seemingly challenging an entire field of well-established research without seeing the benefits of the summative power of this approach.**

**Sharing some of the concerns, we opted for 'growth' broadly as a measure of organism physiological response to climate stressors (and not more specific) for multiple reasons. Disentangling specific growth measurements would dilute our analysis greatly due to the wide disparity in the experimental approaches for measuring growth responses to climate stressors. Furthermore, we wanted to keep in line with the previously cited meta-analyses that we and the reviewer refer to (Krocker et al., 2013; Harvey et al., 2013; Sampaio et al., 2021), who all used growth and not more specific measurements. As such, analysing growth in this manner has become standardised in the synthesis literature for comparable research.**

**Our methods have been updated to reflect a similar point raised by reviewer 1, and the section now better shows that we extracted data from papers based on end-of-century conditions. Additionally, we included a sentence about only using data from experiments which maintained stable conditions throughout. Salinity in our methods has been updated to reflect we opted for 'change' because Salinity can both increase and decrease because of climate change (given location specific factors) in contrast to oxygenation, temperature and acidification. Hence we opted for both increasing and decreasing under the umbrella term of 'change'. We observed negative growth responses for both increased or decreased salinity; hence we felt justified in including salinity data in this manner.  As salinity was not included in previous analyses we considered this an important addition in our analysis for highlighting the variability of climate stressors and the need to explore them, and for identifying potentially novel trends not present in other meta-analyses which focus largely on pH. Text in the discussion has also been updated to better reflect this.**

"the number of available datasets is very low. We have here on several cases the extreme situation where the "meta-analysis" is done on 1 experimental set-up... To me, it is very difficult to compare pH effects based on quite a lot of experimental setups (90), oxygen (18) and salinity (8). It is getting even worse when focusing on the combination of factors (e.g. pH + S = 2 experimental setups = 1 publication). Of course, when considering families separately, the number of experimental results is, in my opinion, way too low to apply any confidence level on the results (except for Mytilidae maybe)."

**We fully agree with the reviewer that for some families, life stages, and combination of stressors the sample size is indeed low, but ultimately this reflects the current state of the experimental literature and is something that we point out as a major conclusion and recommendation for future work. We highlight the abundance of experiments focusing on pH as a literature bias relative to other stressors. We did not perform meta-analysis specifically on one experimental set up. Upon reviewing the code we provide in addition to our data, the reviewer can see that we used 'stressor' as a grouping moderator in the meta-analysis. (Code line 186 families_experiment_counts <-map(list_of_families, function (family) metadata_bivalves |> filter(Family == family) |> group_by(Stressor) |> summarize(count = n()))). This approach does not perform meta-analysis on any stressor individually, but places it within the context of all others, as a work around to biases in experimental data.**

**We recognise that indicating statistical significance on small sample sizes in instances of low data limits the confidence that can be applied to the specific cases, but within the context of the meta-analytical approach our conclusions are still valid. We specify in the paper that we are just working with what is available in the data. Our hope is that by drawing attention to these deficits and biases, it will encourage experimental work on a wider range of bivalves and climate stressors.**

"I have already mentioned that that because of important differences in the protocols used between the different datasets, the low number of datasets in many cases etc.., I believe that the results presented here have a low confidence level, I will push a bit further, and will take 2 examples showing that I even believe using this method in such conditions can lead to false conclusions."

"First of all, L168-169: "The effect of pH + salinity is intermediate between that of the two single stressors, dampening the salinity effect". This assessment is based on the

observation that salinity (8 datasets, increase, decrease of salinity???) has a stronger effect than pH+S (2 datasets). These 2 datasets are from the work of Dickinson (Dickinson et al., Interactive effects of salinity and elevated CO2 levels on juvenile eastern oysters, Crassostrea virginica. Journal of Experimental Biology 215, 29-43 (2012); Dickinson, O. B. Matoo, R. T. Tourek, I. M. Sokolova, E. Beniash, Environmental salinity modulates the effects of elevated CO2 levels on juvenile hard-shell clams, Mercenaria mercenaria. Journal of Experimental Biology 216, 2607-2618 (2013)). On oysters, they found that each factor leads to decreased growth (soft tissue) but that there is no interaction between these factors. On clams, they also showed that low salinity combined with hypercapnia leads to decrease in growth as compared to the growth measured following exposure to one of these stressors in isolation. So, we are far for the finding that a decrease in pH leads to a dampening of salinity effect...."

**Section 4.4 has been updated to reflect concerns around using experimental studies with disparate protocols. Our discussion now includes a paragraph justifying our use of studies with varying experimental protocol, how this is a consistent approach with comparable literature, how our analyses (meta-regression specifically) show that across the studies we derived our effect sizes from for meta-analysis we do not observe significant variations. All giving us confidence that combining disparate studies in this manner is justified and does not weaken our conclusions.**

**We fear our point might have been misunderstood by our choice of language regarding the sentence "The effect of pH + salinity is intermediate between that of the two single stressors, dampening the salinity effect". We have removed refence to dampening through the text, and just point out that its intermediate between the two other stressors. But again, our approach with meta-analysis was not to drill down to specific results, but try and place them within the wider narrative of the collective experimental data we extracted and analyses.**

**While we consider the inclusion of the salinity information important, we do not place large narrative importance on the impacts of salinity due to the low number of experimental studies. Our restructure of the text that referred to "dampening" aids this. The point of our text is to draw attention to the state of the published data, while also emphasising the 'direction of travel' for bivalves as a group based on these data. We like to convey that salinity is a lesser explored stressor but may have interesting interactions with others, e.g., pH, temp, and oxygen.**

"The second example is connected to the main conclusion of the article: L367-368: "We demonstrate that synergistic combinations of stressors (e.g., effects of combined temperature + O2 + pH change) cause greater reductions in bivalve growth then individual stressors.". This assessment is based on 4 experimental setups, but from one single publication (Stevens, C. J. Gobler, Interactive effects of acidification, hypoxia, and thermal stress on growth, respiration, and survival of four North Atlantic bivalves. Marine Ecology Progress Series 604, 143-161 (2018)). They indicated that for 4 species of bivalves, "The combination of low DO and low pH often interacted antagonistically to yield growth rates higher than would be predicted from either individual stressor (...) Elevated temperature and low pH interacted both antagonistically and synergistically, producing outcomes that could not be predicted from the responses to individual stressors. Collectively, this study revealed species- and size-specific vulnerabilities of bivalves to coastal stressors along with unpredicted interactions among those stressors». Again, we are far from a "greater reduction in growth than individual stressors" with a much more complicated story."

**Our conclusion has been restructured to better highlight manuscript novelty, which results support previous work, and the value added to the literature by our research. It also highlights the bias in our data, and how our novel approach can be expanded to other groups threatened by marine climate change.**

---

## Author Response (AR2)

Dear Prof. Steven Bouillon, dear reviewers,

Thank you for asking my opinion on the manuscript "The Clam Before the Storm: A Meta-Analysis Showing the Effect of Combined Climate Change Stressors on Bivalves" by Rachel Kruft Welton and colleagues. I have read the manuscript with interest and provide my assessment and feedback below.

Overall, I think this is a very timely and valuable contribution to the field which fits the scope of the journal. Meta-analyses like the one presented here are relatively uncommon in the field and, as the authors acknowledge, are often biased towards commercially harvested species of mollusks. The manuscript is overall very well written, and the authors clearly introduce the relevance of the study. There is also a clear discussion of the implications of the work.

**We thank the reviewer for their time, and positive comments about our manuscript. We are glad they consider our work both timely and value added to the wider field.**

The biggest disadvantage of this study design is the effect of positive publication bias on the results. The authors do acknowledge that this bias exist (lines 234-236), but the effect of this bias on the results is not further discussed. I understand that the authors cannot avoid this bias, but I would like to see a bit more discussion on how they think it will influence their conclusions.

**We thank the author for highlighting the importance of publication bias in meta-analysis. As the reviewer has already stated, it is an unfortunate result of the data we extracted from the wider literature. We have added a paragraph that further discusses the results of publication bias, how we are confident still in our conclusions despite this, and recommendations for the experimental literature moving forward.**

Minor comments
Line 25: It is not immediately clear what "this assessment" refers to here, to the assessment of the effect on infaunal taxa, the effect on epifaunal taxa or the difference between the two?

**Text updated to clarify we meant the difference between the two**

Line 81: Add "The" or "Our" in front of "understanding".

**Text updated**

Line 96: To the uninitiated reader, the meaning of "granularity" may not be clear. Perhaps the authors can define it here.

**Definition added**

Line 99: Here, I think the authors should either cite in text the 10 and 4 stressors, respectively, or (perhaps preferable) refer to a table where these are listed so the reader immediately knows which parameters are considered in the study.

**Text updated to direct the readers to a table in our results which outline the stressors and their respective permutations**

Line 113: GitHub is not a persistent depository for data. I therefore suggest the authors deposit their data to a repository which has a persistent DOI number to ensure the dataset used specifically as a basis for the outcomes presented here is findable. It is of course a good idea to keep a working version of the dataset on a repository such as GitHub (which can be kept updated) in case the authors or others would like to repeat the analyses later with a bigger/different set of data.

**Data have been additionally uploaded to Zenodo, DOI is now embedded in the text**

Line 118: I think this should read "The publication date of articles collected ranged…"

**Edited**

Lines 127-129: From this section, it is not clear what the authors consider to be "plausible" conditions. The criteria for this should be better explained.

**Text edited to better explain**

Line 138: "is"/"was" is missing between "death" and "often"

**Edited**

Line 144-146: Do the authors mean "fed ad libidum"? "fed" is too vague, as some studies could have food as a limiting factor on growth if the amount of food is not sufficient for optimal growth while other studies do not have this.

**Text updated**

Line 151-152: "Control values for climate stressors for each article were based on authors' determination of control conditions." This statement is too vague. It is not clear to me what the authors mean here.

**Text edited to better explain**

Line 173: Maybe add here that this is a "linear multivariable mixed effects model".

**Edited**

Line 310-311: It is unclear what the authors mean by "or shelf formation of adults from a high pCO2 low pH micro-environment quite different to the surrounding seawater" Please clarify.

**Text edited for clarification**

Line 314: "may be resulted in" should read "could have resulted in" or "maybe resulted in"
**Edited**

Line 322: Here and elsewhere in the manuscript, wherever statements like "fewer experiments" are made, it would be helpful if the authors provide the number of studies/experiments.
**Edited to include the number of experiments**

Line 338: Rephrase to "and this has been hypothesized"
**Edited**

Line 356: Here and elsewhere, I find the term "free swimming" a bit odd. These taxa do not really swim. Although they have been shown to "jump" or "flap" their valves to propel themselves for short distances to escape predation, this would not really be considered swimming and these animals spend most of their lives resting on the substrate. I would avoid this description.

**Description changed throughout the text to 'motile'**